# Lost-in-Distance: Impact of Contextual Proximity on LLM Performance in Graph Tasks

## Abstract

Despite significant advancements, Large Language Models (LLMs) exhibit blind spots that impair their ability to retrieve and process relevant contextual data effectively. We demonstrate that LLM performance in graph tasks with complexities beyond the "needle-in-a-haystack" scenario—where solving the problem requires cross-referencing and reasoning across multiple subproblems *jointly*—is influenced by the proximity of relevant information within the context, a phenomenon we term "lost-in-distance". We examine two fundamental graph tasks: identifying common connections between two nodes and assessing similarity among three nodes, and show that the model's performance in these tasks significantly depends on the relative positioning of common edges. We evaluate three publicly available LLMs—Llama-3-8B, Llama-3-70B, and GPT-4—using various graph encoding techniques that represent graph structures for LLM input. We propose a formulation for the lost-in-distance phenomenon and demonstrate that lost-in-distance and lost-in-the middle phenomenas occur independently. Results indicate that model accuracy can decline by up to 6x as the distance between node connections increases, independent of graph encoding and model size.

## 1 Introduction

Large Language Models (LLMs) have attained an unprecedented level of generality by leveraging scale and attention-based architectures (Kaplan et al., 2020; Vaswani, 2017). These models exhibit remarkable, often superhuman, capabilities across a diverse range of tasks, including language translation, reading comprehension, and question answering (Costa-jussà et al., 2022; Sanh et al., 2021). Additionally, LLMs are increasingly serving as essential and flexible building blocks for various user-facing machine learning and artificial intelligence applications beyond traditional language processing domains, such as recommendation systems (Geng et al., 2022), graph-related tasks (Wang et al., 2024), knowledge bases (AlKhamissi et al., 2022; Petroni et al., 2019), and more. These applications highlight the versatility of LLMs but also expose new challenges in handling domain-specific data encoded as textual input.

Particularly, by leveraging the extensive common knowledge and powerful semantic comprehension abilities of LLMs, recent research has aimed to apply them to tasks related to graph structures (Wang et al., 2024). LLMs are increasingly being adopted for a variety of tasks that involve graph structures, such as planning in robotics (Andreas, 2022), knowledge extraction using knowledge graphs (Shen et al., 2020; Saxena et al., 2020), and multi-hop question answering (Creswell et al., 2022; Fang et al., 2019). For instance, they have been used to guide agents through structured graph-based environments (Huang et al., 2022). Building upon these applications, recent works by Sanford et al. (2024), Perozzi et al. (2024), and Agarwal et al. (2020) have demonstrated that graph tasks can be encoded into textual formats that allow pre-trained LLMs to solve them as out-of-domain tasks. This innovative approach effectively transforms graph problems into a language that LLMs can understand and process.

While LLMs are being expanded in many applications, they suffer from certain blind spots that significantly affect their performance. In particular, how these models process information in their context and retrieve relevant data to solve the task at hand remains an active area of research (Kaddour et al., 2023). Understanding these limitations is crucial for extending context length (Gemini et al., 2023; Xu et al., 2023; Chen et al., 2023) and improving in-context learning (Zhou et al.,

2022; Wei et al., 2023; An et al., 2024). Recent works have shown that the performance of LLMs depends on the location of information in their context. Primarily, Liu et al. (2023) introduces the "lost-in-the-middle" phenomenon, where information placed in the middle of a prompt is less effectively utilized by the model compared to information at the beginning or end, resulting in significant performance degradation when the position of relevant information in the context changes.

Unlike these previous works which mainly focus on shortcomings of LLMs in NLP tasks, e.g. lost-in-the-middle, we focus on the deficiency of these models in tasks beyond natural language processing, specifically solving fundamental graph problems. This area is heavily under-explored and has wide practical applications (Perozzi et al., 2024; Colon-Hernandez et al., 2021; Xie et al., 2023). Since these tasks require understanding graph structure and relationship between objects, they provide us with great insights into model's blind spots. Through our analysis, we provide insight that LLMs not only have blind spots regarding where information exists in the context, but their performance in solving complex tasks also depends on the *relative* position of information within the context.

Particularly, we look into Common Connection and Similarity tasks, which are the main algorithms used as the backbone of many applications such as molecular design (Tan et al., 2023; Xia et al., 2023), social network analysis (Gao et al., 2024), and recommendation systems (Li et al., 2023). For example, these tasks are the main algorithms in "user-user" and "user-item" recommendations in large industry recommendation products (Xie et al., 2023; Huang et al., 2015; Wu et al., 2022). These tasks not only require understanding of subgraph structures but also demand integration of information and reasoning across subgraphs. We demonstrate that strong, publicly available LLMs universally degrade in performance as when relevant pieces of information are distant from each other. Our analysis shows that this effect is present even when one controls for the effects absolute position of the relevant information in the context. To summarize

- For tasks that require cross-subgraph information lookup, such as identifying common connections or measuring similarity, model performance not only degrades due to the "lost-in-the-middle" effect based on the absolute positions but is also affected by the relative distance between pieces of information in the context—a phenomenon we term "lost-in-distance". The further apart the information is, the more the model's performance deteriorates.

- We demonstrate these shortcomings across different graph encoding algorithms and various publicly available LLMs such as Llama-3-8B, Llama-3-70B (Dubey et al., 2024) and GPT-4 (Achiam et al., 2023) indicating a universal limitation in current architectures.

Our findings suggest that current LLMs have inherent limitations in processing contextual information that is not sequentially localized or is widely dispersed within the input. This has significant implications for their application in domains that require complex reasoning over structured data, such as graph analysis.

## 1.1 NOTATIONS AND DEFINITIONS

We define a graph $\mathcal{G} = (\mathcal{V}, \mathcal{E})$, where $\mathcal{V} = \{v_1, v_2, \ldots, v_n\}$ and $\mathcal{E}$ represent the sets of nodes and edges, respectively. If nodes $v_i$ and $v_j$ are directly connected, we denote the edge between them as $e_{ij} \in \mathcal{E}$. The neighbors of node $v_i$ are defined as $\mathcal{N}(v_i) = \{v_k \in \mathcal{V} \mid e_{ik} \in \mathcal{E}\}$. A subgraph associated with node $v_i$ is defined as $\mathcal{G}_{v_i} = (\{v_i\} \cup \mathcal{N}(v_i), \mathcal{E}_{v_i})$, where $\mathcal{E}_{v_i} = \{e_{ij} \mid e_{ij} \in \mathcal{E}, \ v_j \in \mathcal{N}(v_i)\}$.

We define the distance between a common node $v$ within two subgraphs $\mathcal{G}_u$ and $\mathcal{G}_z$ as the number of tokens separating the two occurrences of node $v$ in the context (i.e., the textual representation of the subgraphs). The overall distance between relevant information for common connections between the two subgraphs is defined as the median of all such distances computed for each common node. Throughout the paper, we use $p$ to indicate position and $d$ to indicate distance.

We use accuracy, as defined below, to measure the performance of an LLM model in solving a given task:

$$\text{Accuracy} = \frac{1}{N} \sum_{i=1}^{N} \mathbf{1}_{\{y_i = \hat{y}_i\}} \times 100\%, \tag{1}$$

where $N$ is the total number of samples in the task, and $y_i$ and $\hat{y}_i$ denote the true answer and the model's answer for the $i$th sample, respectively. If the output of the LLM for sample $i$ is degenerate—such as not following instructions or hallucinating—we consider it an incorrect answer, i.e., $y_i \neq \hat{y}_i$.

## 2 GRAPH ENCODING AND GRAPH TASKS

### 2.1 GRAPH ENCODING FOR LLM

Representing graph-structured data as text is an important step in enabling LLMs to understand graph structures and provide accurate answers to questions. Encoding graphs as text involves representing both nodes and edges. Different graph encodings can lead to varying performance of LLMs in graph reasoning tasks (Agarwal et al., 2020; Fatemi et al., 2024; Zhang et al., 2024a). In this work, we encode nodes as integers, where each node is represented by a unique integer, such that $v_i \in \{0, 1, \ldots, n-1\}$.

We experiment with three encoding functions from Fatemi et al. (2024) to encode edges in the graph, investigating whether patterns are consistently observed across different encoding functions. More specifically, we consider the following edge encoding functions:

- **Incident:** Given a source node $v_i$, the edge information for node $v_i$ is encoded as an adjacency list in natural language. For example, "node $v_i$ is connected to nodes $v_j, v_k$".

- **Adjacency:** Given a source node $v_i$ and a target node $v_j$, the edge is encoded as $(v_i, v_j)$.

- **Expert:** Given a source node $v_i$ and a target node $v_j$, the edge is encoded as $v_i \rightarrow v_j$.

Since the graph tasks considered in this paper only require access to the subgraph and the subgraph structure, we encode only the edge information for the nodes of interest. This is a common practice where a subgraph is extracted from a database before being processed by a compute engine (Shao et al., 2013). Figure 1 shows an example about only including a subgraph with three encoding functions in the prompt. In this example, node 0 and node 1 are nodes of interest so we only encode their subgraph in the prompt.

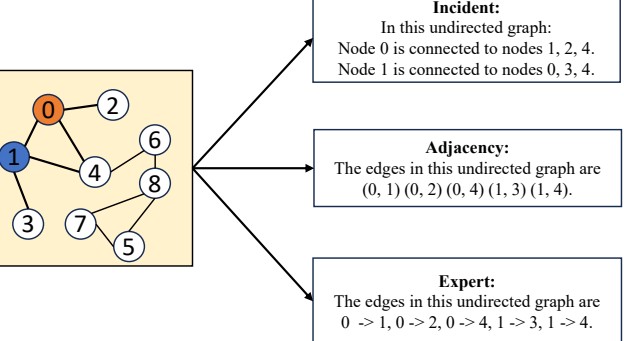

Figure 1: Three graph encoding functions, with node 0 and node 1 serving as the nodes of interest. The figure is inspired by Fatemi et al. (2024).

### 2.2 GRAPH GENERATION

In this paper, we build upon previous studies (Huang et al., 2022; Fatemi et al., 2024; Zhang et al., 2024b) by conducting experiments on randomly generated graphs. We utilize the Erdős–Rényi (ER) graph generator (Erdős & Rényi, 1959) to create undirected graphs. We experiment with relatively large graphs comprising $n = 1000$ nodes. The undirected edge $e_{ij}$ between nodes $v_i$ and $v_j$ is generated with probability $P(e_{ij} \in \mathcal{E})$. We set $P(e_{ij} \in \mathcal{E}) = 0.1$ throughout the main manuscript, and results for other values of $P(e_{ij} \in \mathcal{E})$ are presented in the Appendix for brevity.

## 2.3 GRAPH TASKS

We aim to analyze the performance of LLMs in three fundamental graph problems which require models to have thorough understanding of the input graph structure.

1. **Edge Existence:** Given two nodes $v_i$ and $v_j$ sampled from a graph $\mathcal{G}$, node $v_i$ and node $v_j$ are directly connected if $e_{ij} \in \mathcal{E}$. The edge existence task is to ask LLMs whether node $v_i$ and node $v_j$ are directly connected.

2. **Common Connection:** Given two nodes $v_i$ and $v_j$ sampled from a graph $\mathcal{G}$, the common connection between two nodes are $\mathcal{N}(v_i) \cap \mathcal{N}(v_j)$. For this task, we ask LLMs to find the number of common connections between node $v_i$ and node $v_j$, denoted as $|\mathcal{N}(v_i) \cap \mathcal{N}(v_j)|$.

3. **Similarity:** Given three nodes $v_i$, $v_j$ and $v_k$ sampled from a graph $\mathcal{G}$, we let $v_j$ be the source node and $v_i$ and $v_k$ be the target nodes. The task for LLMs is to compare the number of common connections $|\mathcal{N}(v_i) \cap \mathcal{N}(v_j)|$ and $|\mathcal{N}(v_j) \cap \mathcal{N}(v_k)|$.

Note that these tasks are roughly ordered in terms of general complexity. For example, solving the edge existence only depends on the model being able to retrieve the edge information from the representation. One step further, in finding the number of common connections, models needs to first identify the set of shared connections between two nodes and then calculate the size of that set. Finally, the similarity task is more complex than the common connection task, as it requires LLMs to consider *three* nodes and identify two sets of common connections and then compare their sizes. As a result, these tasks are a good representative set to evaluate LLMs since they require LLMs to both retrieve and reason about the graph information. Furthermore, these tasks are also essential and the building blocks for solving practical problems in applications such as recommendation systems (Ying et al., 2018), protein folding (Strokach et al., 2020), bad actor detection (Papegnies et al., 2017) or any other task that requires graph understanding.

## 3 LOST-IN-THE-MIDDLE FOR EDGE EXISTENCE

The edge existence task is analogous to the needle-in-a-haystack problem (Ivgi et al., 2023) and the document question-answering task (Liu et al., 2023), as it requires the LLM to retrieve the answer from the prompt without performing any computation. Building upon prior work in the literature by Liu et al. (2023), this study demonstrates the impact of the position of relevant information on the performance of LLMs. Specifically, it is shown that the accuracy in the edge existence task decreases when the information about the edge in question is placed in the middle of the prompt.

The prompt structure is constructed using the following procedure, which enables controlling the location of information within the prompt:

1. Randomly sample two nodes from a graph along with their corresponding connections.

2. Randomly select nine additional nodes as noise nodes and incorporate their textual subgraph encodings into the prompt. This step is necessary to examine the impact of the position of relevant information.

3. Group the subgraph structures of the two nodes of interest and position them at the beginning, middle, or end of the input context.

4. Query the model to determine whether an edge exists between the two nodes of interest.

An example of a prompt with different positions for the two nodes of interest is illustrated in Figure 2.

### 3.1 EXPERIMENTAL RESULTS

**Lost-in-the-middle can happen in the edge existence task.** To demonstrate the lost-in-the-middle phenomena in edge existence task, we experiment with the state of the art model as of writing this paper GPT-4. The experiment results are averaged over twenty randomly generated graph where from each graph we randomly select two nodes and form the edge existence prompt as described in previous section.

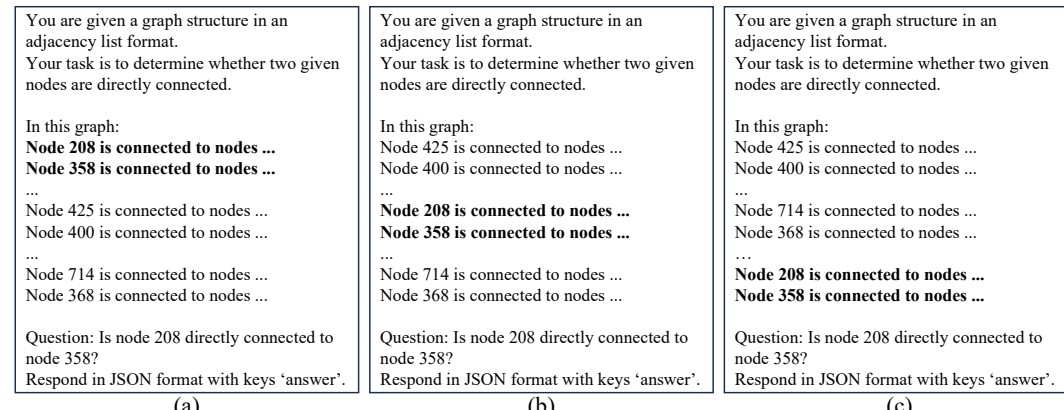

Figure 2: Example of the edge existence task, illustrating the placement of the nodes of interest subgraph (nodes 208 and 358) at (a) the beginning, (b) the middle, and (c) the end of the graph structure.

Figure 3 shows that all encodings can cause the LLM to lose the information in the middle of the prompt. The best performance occurs when the relevant information is either at the beginning or the end of the entire subgraph structure. Even for the incident encoding which has the best performance among all encodings, the LLM still has the worst performance when the answer is located in the middle of the prompt.

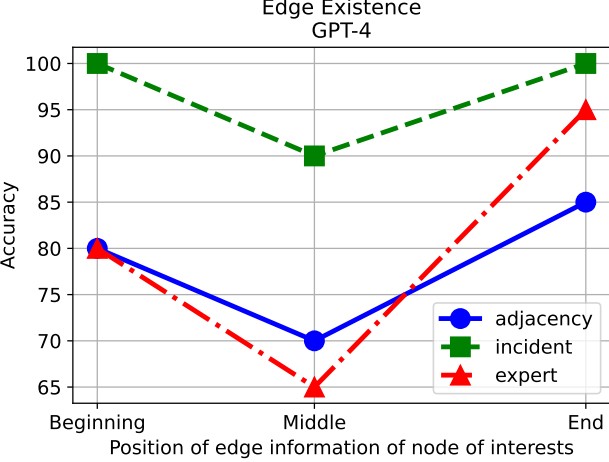

Figure 3: The effect of the position of the relevant information on the edge existence task.

# 4 LOST-IN-DISTANCE

Tasks such as the edge existence require LLMs to perform needle-in-a-haystack retrieval, which, as previously shown, suffers from the lost-in-the-middle phenomenon in long contexts. However, in many tasks, the model not only needs to look up relevant information in the context but also requires to perform cross-referencing between retrieved information. For example, tasks like the common connection require the model to retrieve connections that *jointly* appear in both subgraphs.

We hypothesize that for tasks requiring cross-referenced retrieval, the model's performance is also impacted by the distance between relevant pieces of information, a phenomenon we term lost-in-distance. Specifically, for these tasks, the model's performance is influenced by two compounding

phenomena: lost-in-the-middle when retrieving relevant information and lost-in-distance when performing a join between retrieved information.

To explore this, we define $G(p)$ as the model's performance when the relevant information is at position $p$. Similarly, we define $F(p_1, p_2)$ as the model's performance when the relevant information is at positions $(p_1, p_2)$. The value of $F(p_1, p_2)$ is estimated based on the accuracy of model in a complex task that requires cross-referencing. We hypothesize that $F$ and $G$ have the following relationship:

$$F(p_1, p_2) = \gamma \, G(p_1) \, G(p_2) \, H(d), \tag{2}$$

where $d = |p_2 - p_1|$ represents the distance between relevant information in the prompt and $H(d)$ represents the effect of lost-in-distance.

In the experimental section, by studying LLM performance on common connection and similarity tasks, we first demonstrate that lost-in-the-middle alone cannot explain the model's performance degradation in solving tasks that require joint reasoning across multiple subgraphs, and that it is affected by another factor, lost-in-distance. Then, by leveraging the experimental results, we measure the goodness of fit for Equation 2 in Section 6.

## 5 EXPERIMENTATION

In our initial experiments, we focused on the common connection task. This task requires the model to determine the number of common connections between two nodes by joining information across two subgraphs. Our results demonstrate that the models' performance degrades as the distance between the relevant pieces of information in the two subgraphs increases. Specifically, when the information about each node's connections is placed further apart in the context, the models struggle to effectively retrieve and integrate this information to compute the correct number of common connections.

We then investigated how the lost-in-distance impacts tasks that require multiple cross-referencing steps, such as the similarity task. In the similarity task, the model needs to first identify the common connections between each of the two nodes and a reference node, and then compare these sets to determine the degree of similarity. Our findings reveal that performance degradation is even more pronounced in this case, as the task requires the model to perform multiple join operations over dispersed pieces of information within the context.

### 5.1 EXPERIMENTAL SETUP

Leveraging in-context learning (Dong et al., 2022; Wei et al., 2023), we conducted experiments using both closed-source models (GPT-4) and open-source models (Llama-3-8B-Instruct and Llama-3-70B-Instruct). For all models, we set the decoding temperature to zero to ensure the generation of deterministic answers. In each sample, we randomly selected two or three nodes as the nodes of interest for the common connection and similarity tasks, respectively. We performed experiments on hundreds of thousands of randomly generated graphs to draw statistically significant conclusions regarding LLM behavior. The experimental results were then averaged across multiple samples.

### 5.2 COMMON CONNECTION TASK

In this section, we demonstrate the effect of increased distance on solving the common connection task. To create an input prompt for this task and to control the relative distance of relevant information (common neighbors), we use the following methodology:

1. Sample two nodes from a given graph and extract their corresponding subgraphs.

2. Within each subgraph, group the common connections.

3. Within each subgraph, position the common connections at the beginning, middle, or end of the textual encoding of the subgraph.

The above recipe, specifically the grouping of relevant information into three positions—beginning, middle, and end (as illustrated in Figure 4 for adjacency encoding)—enables us to control the relative distance between common connections within the prompt. This allows us to investigate the effects on the model's performance when the relative distance is small, medium, or large. We denote the positions of relevant information within the first and second subgraphs as $(p_1, p_2)$, where $p_1 \in \{0, 1, 2\}$ and $p_2 \in \{3, 4, 5\}$, respectively, for the sake of brevity.

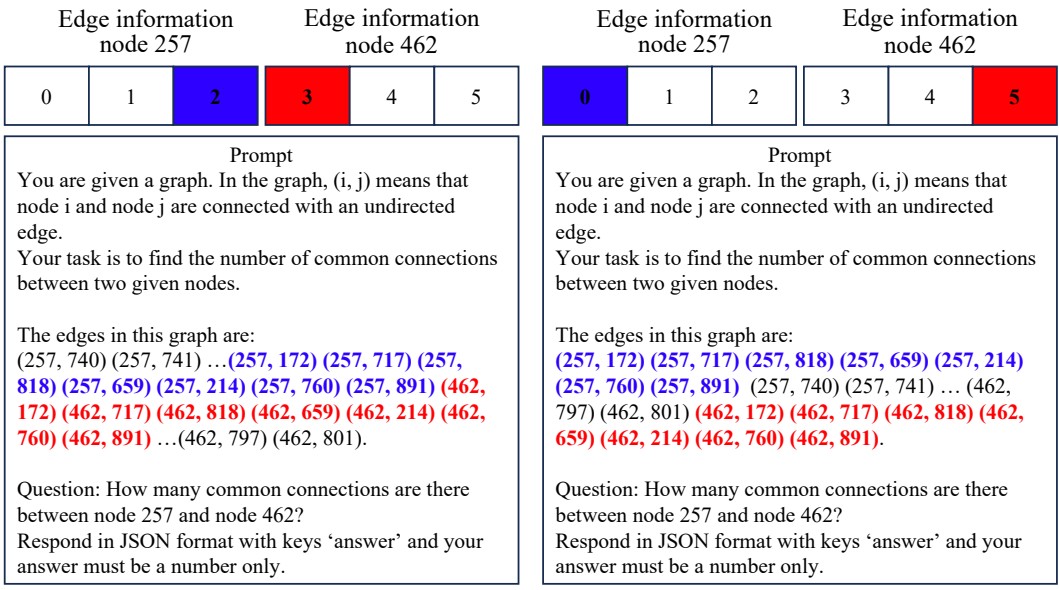

Figure 4: An example illustrating the placement of relevant information, highlighted in blue and red, at different positions using the adjacency encoding function for the common connection task. Relevant information is grouped at positions $0$, $1$, or $2$ within the first node's (node $257$) subgraph and at positions $3$, $4$, or $5$ within the second node's (node $462$) subgraph. The left plot depicts the smallest distance between relevant information, while the right plot shows the largest distance.

### 5.2.1 LOST-IN-DISTANCE IN COMMON CONNECTION TASK

The results presented in Figure 5 illustrate the impact of varying the positions of common edges within each subgraph (following the methodology outlined in the previous section) on the model's performance in the common connection task. Unlike the edge existence task, the model's performance is influenced not only by the lost-in-the-middle phenomenon but also by the relative distance between common connections.

With the position of relevant information fixed in one subgraph, we observe that the model's performance degrades when the other subgraph is positioned closer to the middle of the prompt, influenced by the lost-in-the-middle phenomenon. For example, in adjacency encodings (Figure 5, middle plot), when the first node's common connection is at position $0$ (the beginning of the prompt), the model's performance deteriorates from 40% to 20% as the second node's common connection shifts from position $5$ (the end) to position $3$ (the middle). However, in contrast to the lost-in-the-middle phenomenon, Figure 5 demonstrates that across all three graph encodings, the model achieves optimal performance when relevant information is centrally located, with minimal distance between components at positions $(2, 3)$. This illustrates the effect of lost-in-distance. Furthermore, when the first node's common connection is at position $2$, the model's accuracy drops by up to 50% as the second node's common connection shifts from position $3$ (the middle of the prompt) to position $5$ (the end of the prompt), thereby increasing the distance between relevant information. These observations confirm that the lost-in-distance phenomenon and the lost-in-the-middle effect have independent, compounding effects on model performance, as hypothesized in Equation 2.

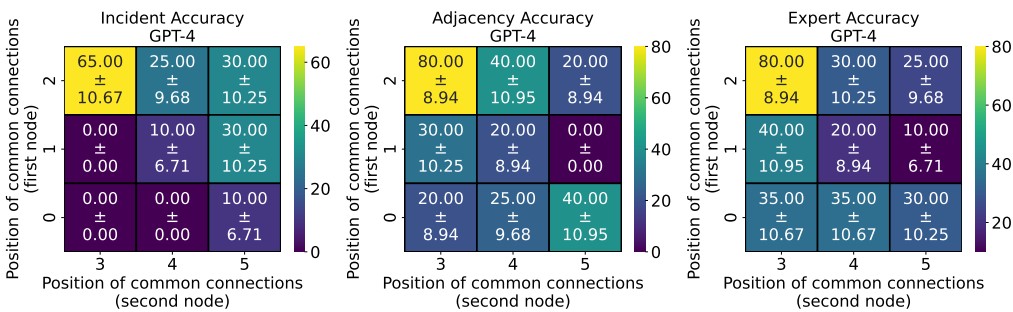

Figure 5: The effect of lost-in-distance on the common connection task. The number in each block is accuracy ± standard deviation.

Table 1: Thresholds of each distance group for three graph encoding functions where distance is measured in number of tokens.

| Graph Encoding | Small Distance | Medium Distance | Large Distance |
|---|---|---|---|
| Incident | $\leq 219$ | $219 \sim 399$ | $> 399$ |
| Adjacency | $\leq 425$ | $425 \sim 785$ | $> 785$ |
| Expert | $\leq 354$ | $354 \sim 654$ | $> 654$ |

## 5.3 SIMILARITY TASK

Solving the similarity between three nodes $v_i$, $v_j$, and $v_k$ requires the model to perform two common connection tasks, $|\mathcal{N}(v_i) \cap \mathcal{N}(v_j)|$ and $|\mathcal{N}(v_j) \cap \mathcal{N}(v_k)|$, and subsequently compare the results. As a result, the model needs to execute two cross-referencing operations between the subgraphs: one between $\mathcal{G}_{v_i}$ and $\mathcal{G}_{v_j}$, and the other between $\mathcal{G}_{v_j}$ and $\mathcal{G}_{v_k}$. Therefore, as we will demonstrate in this section, solving the similarity task inherently suffers from the lost-in-distance phenomenon.

To measure the effect of the lost-in-distance phenomenon, we select three nodes—$v_i$, $v_j$, and $v_k$—from a given graph and randomly shuffle the edges within each node's subgraph. We designate $v_j$ as the source node for the similarity task, $v_i$ as the first target node, and $v_k$ as the second target node. In all scenarios, to mitigate the influence of the lost-in-the-middle effect and highlight the effect of lost-in-distance, the textual encoding of the source node $v_j$'s subgraph ($\mathcal{G}_{v_j}$) is positioned at the center of the prompt, while the subgraphs of the other two nodes are placed one before and one after it.

We quantify the lost-in-distance effect by calculating the median distance, measured in terms of tokens, between the common connections of the two subgraphs, specifically $|\mathcal{N}(v_i) \cap \mathcal{N}(v_j)|$ and $|\mathcal{N}(v_j) \cap \mathcal{N}(v_k)|$. The distance distribution is illustrated in Figure 6 for three different graph encodings. We utilize the thresholds presented in Table 1 to categorize the distances into small, medium, and large groups. Furthermore, in order to make sure more uniform coverage, we employ rejection sampling to ensure that each distance group contains one hundred samples with balanced responses.

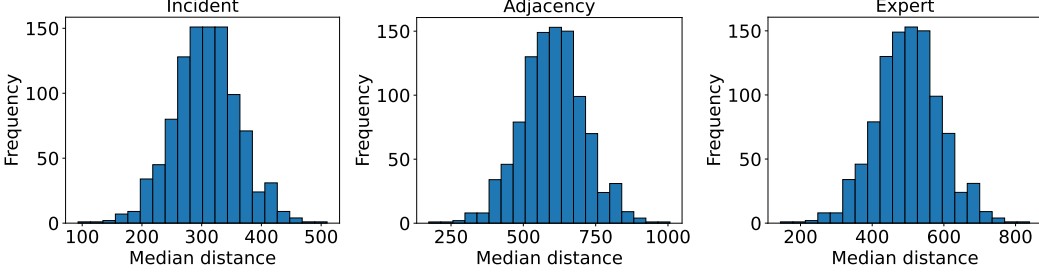

Figure 6: The distribution of median distance, in number of tokens, for three graph encoding functions.

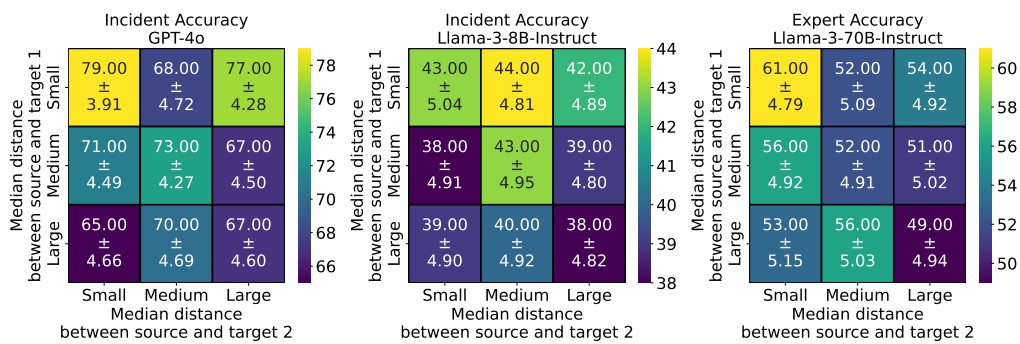

Figure 7: The effect of lost-in-distance on the similarity task is illustrated. As the distance between target node 1 and target node 2 increases, the model's performance degrades accordingly. The numbers in each block represent the accuracy ± standard deviation obtained through bootstrap sampling.

To eliminate potential biases, for the three subgraphs $\mathcal{G}_{v_i}$, $\mathcal{G}_{v_j}$, and $\mathcal{G}_{v_k}$, where $v_j$ is the source node for similarity, we generate questions randomly chosen from the following two templates:

- *Is the number of common connections between **node $v_j$ and node $v_k$** greater than the number of common connections between **node $v_i$ and node $v_j$**?*

- *Is the number of common connections between **node $v_i$ and node $v_j$** greater than the number of common connections between **node $v_j$ and node $v_k$**?*

An example of the prompt for the similarity task is presented in Appendix C.1. Since the similarity task inherently involves solving two common connection tasks and a comparison task, we adopt Chain-of-Thought prompting (Wei et al., 2022) to guide the model in solving the task step by step, thereby obtaining the most accurate answers. Detailed descriptions of the prompts and further analysis of the LLMs' failure rates in following CoT instructions, along with examples of answer degeneration, are provided in Appendix A.2. GPT-4 and Llama-3-70B exhibit low failure rates, while Llama-3-8B demonstrates a high failure rate across all graph encodings, including a 69% failure rate in expert encodings.

### 5.3.1 LOST-IN-DISTANCE IN SIMILARITY TASK

For brevity, we present the results of one encoding for each model in Figure 7, with all results summarized in Appendix C.2. Our findings indicate that when both distances are minimal, GPT-4 and Llama-3-70B-Instruct exhibit the best performance. Llama-3-8B-Instruct, which has a high failure rate in following instructions as described in Appendix A.2, demonstrates the second-best performance, though it is not significantly different from the top performers.

Specifically, performance at the largest distances is significantly worse compared to that at the smallest distances. As the distances increase (i.e., along the diagonal elements), the performance of all models deteriorates. In Llama-3-70B, we observe a 12% drop in model accuracy when the distance between common connections for both $|\mathcal{N}(v_i) \cap \mathcal{N}(v_j)|$ and $|\mathcal{N}(v_j) \cap \mathcal{N}(v_k)|$ increases, shifting from the (Small, Small) index to the (Large, Large) index in the heatmap plot. These results highlight that the lost-in-distance phenomenon adversely affects model performance in similarity tasks.

## 6 GOODNESS OF FIT FOR LOST-IN-DISTANCE

In this section, we employ the Equation 2 function to capture the effects of the lost-in-distance phenomenon and to separate its impact from that of the lost-in-the-middle effect. To evaluate the goodness of fit for the lost-in-distance function defined in Equation 2, we compare it to a simpler function that accounts solely for the lost-in-the-middle effect as follows:

$$\mathbb{E}[F(p_1, p_2)|G(p_1), G(p_2)] = \gamma G(p_1)G(p_2), \tag{3}$$

$$\mathbb{E}[F(p_1, p_2)|\gamma, G(p_1), G(p_2)] = \gamma G(p_1)G(p_2)H(|p_2 - p_1|), \tag{4}$$

Table 2: RMSE for lost-in-the-middle only and Equation 2 functions with three encodings.

| Encoding | Model | RMSE (train) | RMSE (test) |
|---|---|---|---|
| Incident | Lost-in-the-middle only | 22.42 | 27.09 |
| | Equation 2 | 10.04 | 14.50 |
| Adjacency | Lost-in-the-middle only | 21.43 | 22.36 |
| | Equation 2 | 15.16 | 13.61 |
| Expert | Lost-in-the-middle only | 24.50 | 26.69 |
| | Equation 2 | 12.84 | 17.42 |

where $H(|p_2 - p_1|))$ is the effect of lost-in-distance $d = |p_2 - p_1|$.

To measure the goodness of fit we leverage results and output of common connection experiments but the result and findings here are extendable to similarity task as well. We randomly split samples into training and test sets of equal size. Using the training set, we first estimate $\hat{G}(\cdot)$ using interpolation based on the accuracy observed in the edge existence task. Then, we estimate $\gamma$ by regressing $F(p_1, p_2)$ onto $\hat{G}(p_1)\hat{G}(p_2)$. Finally, given the estimated $\hat{\gamma}$ and $\hat{G}(\cdot)$, we estimate $H(\cdot)$ using

$$\hat{H}(d) = \frac{1}{|\mathcal{D}_d|} \sum_{(p_1, p_2) \in \mathcal{D}_d} \frac{F(p_1, p_2)}{\hat{\gamma}\hat{G}(p_1)\hat{G}(p_2)}, \tag{5}$$

where $\mathcal{D}_d = \{(p_1, p_2) | |p_2 - p_1| = d\}$.

We evaluate the goodness of fit for both functions using the root mean squared error (RMSE) between the predicted and observed accuracy in the test set.

Table 2 shows that Equation 2 function, which includes the lost-in-distance effect, has a smaller RMSE compared to the lost-in-the-middle only function. Moreover, $\hat{H}(\cdot)$ in Figure 8 indicates that smaller distance results in better performance, up to 3x, after accounting for the lost-in-the-middle effect.

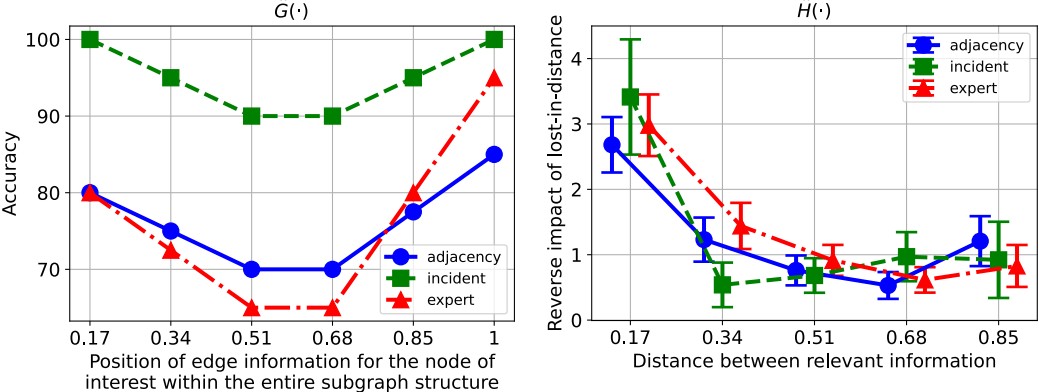

Figure 8: The left plot is $\hat{G}(\cdot)$ and the right plot is $\hat{H}(\cdot)$. To better visualize the error bars in $\hat{H}(\cdot)$ estimation, we slightly shift the adjacency encoding to the left and the expert encoding to the right. Numbers in x-axis are normalized to the prompt length.

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

# A  ANALYSIS

## A.1  CONTEXT LENGTH

Table 3 summarizes the average context length (i.e., the number of tokens) for each task and each graph encoding. We use the tokenizer of Llama-3 to calculate the context length for Llama-3-8B-Instruct and Llama-3-70B-Instruct and use the tiktoken library (OpenAI, 2023b) to calculate the context length for GPT-4 and GPT-4o. The incident encoding produces the shortest context length, while the adjacency encoding results in the longest context length.

Table 3: Input and output context length in each encoding and each task.

| Graph Task | Graph Encoding | Average Input Length | Average Output Length | | |
|---|---|---|---|---|---|
| | | | GPT-4/4o | Llama-3-8B-Instruct | Llama-3-70B-Instruct |
| | Incident | 3409.50 | 13 | 8 | 7 |
| Edge Existence | Adjacency | 6598.10 | 13 | 8 | 7 |
| | Expert | 5514.75 | 13 | 8 | 7 |
| | Incident | 662.55 | 13 | 8 | 7 |
| Common Connection | Adjacency | 1261.10 | 13 | 8 | 7 |
| | Expert | 1068.25 | 13 | 8 | 7 |
| | Incident | 1263.37 | 153.46 | 695.31 | 91.76 |
| Similarity | Adjacency | 2164.75 | 142.32 | 1126.30 | 106.86 |
| | Expert | 1869.29 | 140.34 | 1453.97 | 97.73 |

## A.2  ANSWER DEGENERATION

LLMs sometimes fail to follow instructions and generate responses that do not adhere to the expected output template. We classify these degenerate responses as incorrect answers, i.e., $y_i \neq \hat{y}_i$ for accuracy calculation in Equation 1. Llama-3-8B-Instruct is less likely to generate a final answer compared to GPT-4o and Llama-3-70B-Instruct, which explains why it has lowest accuracy in Figure 7. Table A.2 summarizes the percentage of samples in which models fail to follow instructions. Generally, the most common patterns of degenerate answers are as follows:

- **Repetition**: LLMs sometimes repeat the same context until they reach the maximum number of output tokens, thereby failing to generate a final answer.

- **Self-contradiction**: In CoT prompting, LLMs are asked to answer the main question based on their responses to subquestions. However, we find that LLMs sometimes provide an incorrect final conclusion. For example, as shown in Figure 9 where we ask the LLM "*is the number of common connections between **node 658 and node 535** greater than the number of common connections between **node 535 and node 807**?*", the LLM determines that the number of common connections between node $658$ and node $535$ is 6, and between node $535$ and node $807$ is $4$, but the final answer is "no" when it should be "yes".

---

**Model Answer**

<sub a>The unique common connections between node 658 and node 535 are 30, 346, 575, 605, 910, and 642.</sub a>

<sub a>6</sub a>

<sub a>The unique common connections between node 535 and node 807 are 303, 633, 642, and 92.</sub a>

<sub a>4</sub a>

<final a>No</final a>

---

Figure 9: Example where the answer is self-contradictory. The question in this example is that "*is the number of common connections between **node 658 and node 535** greater than the number of common connections between **node 535 and node 807**?*".

Table 4: Percentage of samples where each model generates degenerated answer.

| Graph Encoding | Percentage of not generating the answer | | |
|---|---|---|---|
| | GPT-4o | Llama-3-8B-Instruct | Llama-3-70B-Instruct |
| Incident | 0.78% | 18.78% | 0.11% |
| Adjacency | 4.11% | 61.11% | 0.22% |
| Expert | 4.44% | 69.00% | 0.22% |

## B  COMMON CONNECTION: MORE RESULT

Figure 10 illustrates the impact of the lost-in-distance phenomenon on the GPT-4o model (OpenAI, 2023a) in solving the common connection task. Although the accuracy metrics slightly differ from those in Figure 5 for GPT-4, the pattern of the lost-in-distance effect remains consistent.

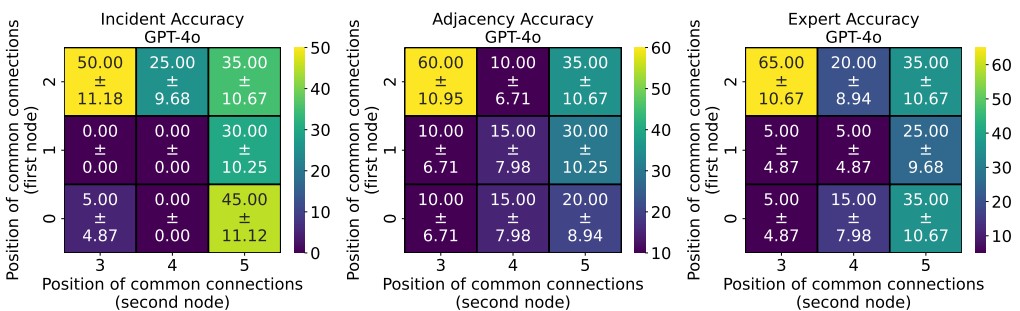

Figure 10: The effect of the position of the relevant information on the GPT-4o model solving the common connection task.

## C  SIMILARITY TASK

### C.1  PROMPT EXAMPLE

Figure 11 illustrates an example of the similarity task prompt, as described in Section 5.3, along with GPT-4o's answer for solving the similarity task using incident graph encoding.

### C.2  ALL RESULTS

Figure 12 presents the results of the similarity task at a density of 0.1 across three models (GPT-4o, Llama-8B, Llama-70B) and three different graph encodings. For all models utilizing the graph encoding functions, we observe the typical lost-in-distance pattern, where performance at the $(\mathrm{Small}, \mathrm{Small})$ index is better than at the $(\mathrm{Large}, \mathrm{Large})$ index.

## D  EFFECT OF GRAPH DENSITY

The lost-in-distance effect remains consistent across different graph densities, i.e., different values of $P(e_{ij} \in \mathcal{E})$ in Erdős–Rényi (ER) randomly generated graphs. Graph density affects the input sequence length in a linear manner; higher densities result in proportionally longer input sequences, as demonstrated in Table 5.

Figure 13 illustrates that increasing the context length by raising graph density follows the same pattern of the lost-in-distance effect in similarity tasks. Specifically, accuracy declines progressively from top to bottom and left to right as the distances between common edges within each subgraph increase. Additionally, the figure demonstrates that for smaller context lengths, corresponding to graphs with low density, the results are noisier and the effect of lost-in-distance diminishes.

---

**Prompt**

You are given an undirected graph structure in an adjacency list format.

Given three nodes i, j, and k, your task is to determine if the number of common connections between node i and node j is greater than the number of common connections between node j and node k.

You MUST first answer several subquestions in '<sub q></sub q>' tags in order and put your answers in '<sub a></sub a>' tags in order. Finally, answer the question in the '<final q></final q>' tag based on the given graph and your answers to the subquestions. Put your final answer in the '<final a></final a>' tag.

In this graph:
Node 963 is connected to nodes {neighbors of node 963}.
Node 452 is connected to nodes {neighbors of node 452}.
Node 423 is connected to nodes {neighbors of node 423}.

<sub q>What are the unique common connections between node 963 and node 452? Your answer MUST start with 'The unique common connections between node 963 and node 452 are'.</sub q>
<sub q>How many common connections are there between node 963 and node 452? Your answer MUST be a number only.</sub q>
<sub q>What are the unique common connections between node 452 and node 423? Your answer MUST start with 'The unique common connections between node 452 and node 423 are'.</sub q>
<sub q>How many common connections are there between node 452 and node 423? Your answer MUST be a number only.</sub q>
<final q>Based on the given graph and previous answers, is the number of common connections between node 963 and node 452 greater than the number of common connections between node 452 and node 423? Your answer MUST be 'Yes' or 'No' only.</final q>

---

**Model Answer**

<sub a>The unique common connections between node 963 and node 452 are 704, 799, 972, 430, 827, 531, 628, 496, 551, 908, 104.</sub a>
<sub a>11</sub a>
<sub a>The unique common connections between node 452 and node 423 are 104, 832, 496, 906, 824, 470, 550, 766, 634, 740.</sub a>
<sub a>10</sub a>
<final a>Yes</final a>

Figure 11: Example of the CoT prompting for the similarity task.

Table 5: Average Token Length from different Graph Density, $P(e_{ij} \in \mathcal{E})$, for ER graphs using expert encoding

| Graph Density | Average Token Length |
|---|---|
| 0.05 | 1145 |
| 0.10 | 1869 |
| 0.15 | 2633 |

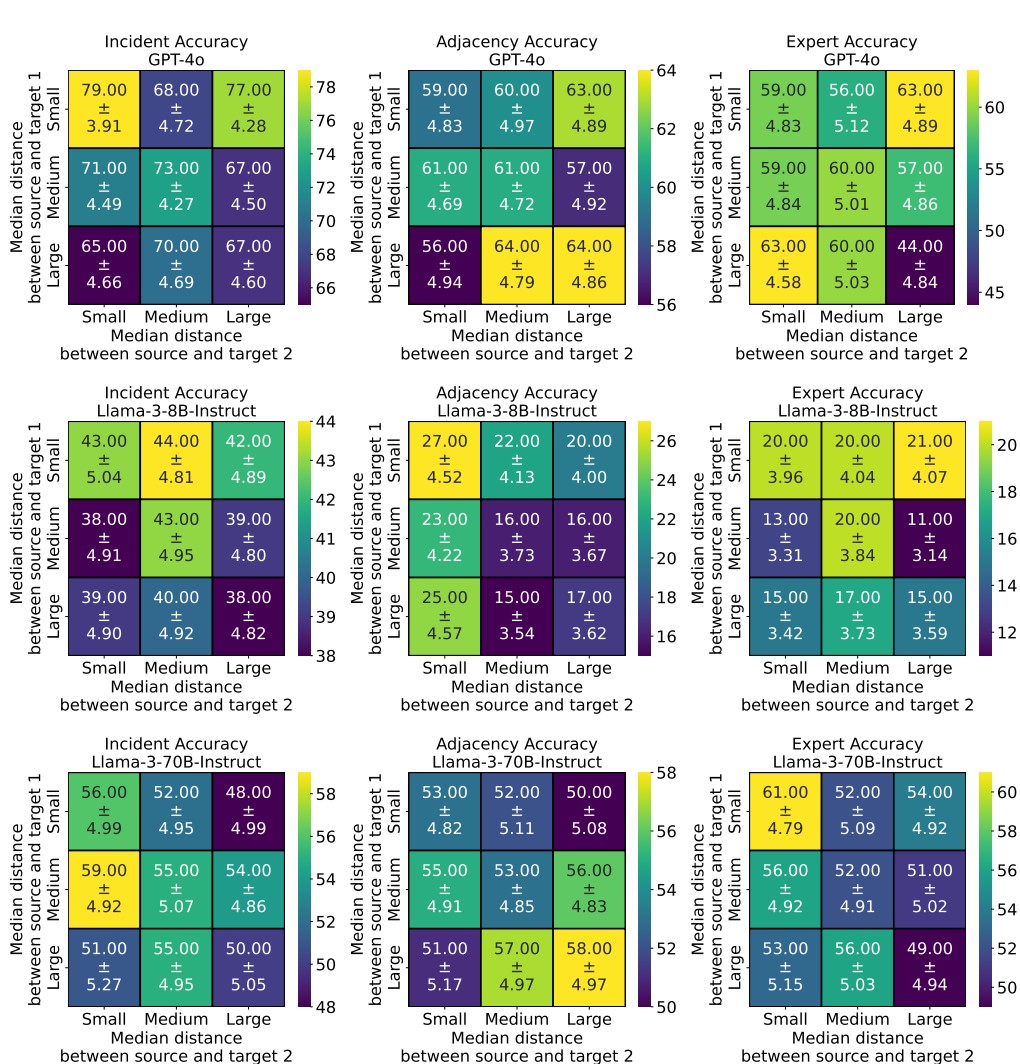

Figure 12: All results in the similarity task with density = 0.1.

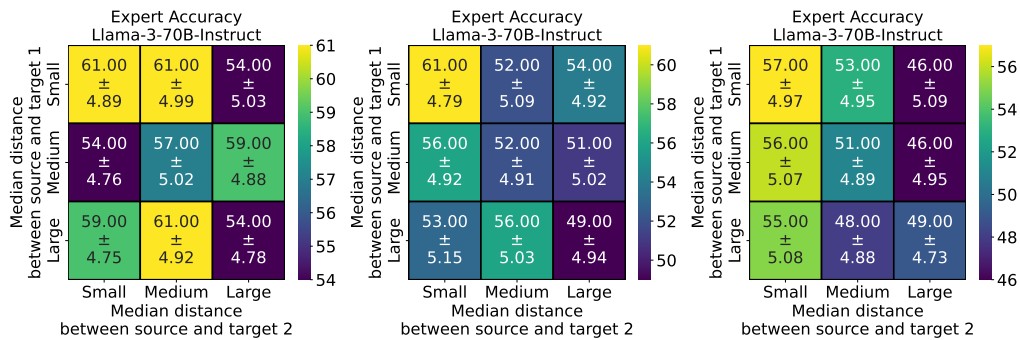

Figure 13: Results from similarity tasks with three different density values, $P(e_{ij} \in \mathcal{E})$, (left) 0.05 , (middle) 0.1, and (right) 0.15.

