# OpenReview forum: "Lost-in-Distance: Impact of Contextual Proximity on LLM Performance in Graph Tasks"
_ICLR.cc/2025/Conference — Submitted to ICLR 2025_

### Official Review · Reviewer_meno · 2024-11-02

**Soundness:** 2
**Presentation:** 2
**Contribution:** 1
**Rating:** 3
**Confidence:** 4

**Summary:**

This paper explores a notable phenomenon in LLMs regarding their performance on complex graph tasks, specifically how effectiveness decreases as the distance between relevant contextual information increases. The authors introduce the concept of "lost-in-distance," investigating two fundamental graph tasks: identifying common connections between two nodes and assessing similarity among three nodes. The study conducts experiments on three publicly available LLMs, utilizing various graph encoding techniques to represent graph structures.

**Strengths:**

1. The problem in this paper is novel, introducing the concept of "lost-in-distance" in the context of LLMs working on complex graph tasks. This perspective offers an innovative angle on understanding LLM performance.
2. The study includes experiments on both closed-source and open-source models, enhancing the robustness of the findings. The authors utilize three different encoding functions to transform graph data into text formats suitable for LLM processing, showcasing a comprehensive approach to graph encoding.

**Weaknesses:**

1. The problem investigated in this paper may not be a critical aspect of LLM reasoning processes. There are already several methods that can significantly enhance LLMs' reasoning capabilities for complex graph-related problems; however, the authors do not discuss these important related works, such as methods for improving LLMs' inference capabilities on knowledge graphs [1] and Graph Retrieval-Augmented Generation [2].
2. The authors do not provide an in-depth theoretical analysis of why LLMs exhibit the lost-in-distance phenomenon in complex graph tasks.
3. The paper identifies a pertinent issue but does not provide corresponding solutions or model innovations. It lacks a discussion of potential measures or strategies to address this phenomenon, which could have significantly enhanced the overall contribution of the research.
4. I do not believe this paper is complete; it lacks a thorough discussion of related work and does not provide a conclusion section.

[1] Sun J, Xu C, Tang L, et al. Think-on-graph: Deep and responsible reasoning of large language model with knowledge graph[J]. arXiv preprint arXiv:2307.07697, 2023.
[2] Edge D, Trinh H, Cheng N, et al. From local to global: A graph rag approach to query-focused summarization[J]. arXiv preprint arXiv:2404.16130, 2024.

**Questions:**

Does the "lost-in-distance" phenomenon appear in real-world application scenarios, such as knowledge graph question answering or recommender systems? Are there any experiments that provide evidence for this?

---

### Official Review · Reviewer_zXbg · 2024-11-03

**Soundness:** 2
**Presentation:** 2
**Contribution:** 2
**Rating:** 3
**Confidence:** 3

**Summary:**

Motivated by the prevalent phenomenon "lost-in-the-middle" of LLMs solving general tasks, this paper investigates how LLM performance changes for graph tasks when relevant information is positioned differently ("lost-in-middle"), and when pieces of relevant information are provided at varying distances ("lost-in-distance"). Focusing on Edge Existence, Common Connection, and Similarity tasks, the empirical findings demonstrate that LLMs do exhibit the lost-in-distance, and that changing the position of relevant information can significantly enhance performance.

**Strengths:**

This paper **studies an important yet unexplored research question**: How does LLMs' performance change when relevant information is given in different positions within the query?

Personally, I believe understanding this problem can reveal insights into LLMs' reasoning mechanism for graph tasks, unlocking their potentials to tackle graph-related tasks more intelligently.

**Weaknesses:**

This paper is more like an experimental report than a research paper. I will elaborate on my opinion as follows:

1. The primary focus of this paper is to reveal that LLMs' performance on graph tasks is significantly influenced by the positioning of relevant information in the query and the distance between relevant pieces of information. However, **some conclusions are weakened by limited empirical investigations**:
   * (a) The experimented LLMs are limited, e.g., for showing "lost-in-the-middle" effect, only GPT-4 is considered.
   * (b) The number of tested samples is limited, e.g., only 20 cases for "lost-in-the-middle", and an unspecified number for remaining experiments.
   * (c) Lack of evaluation on existing graph reasoning benchmarks [1, 2], and the synthesized data is overly simplistic, e.g., only 9 additional noisy nodes for "lost-in-the-middle".

2. The authors primarily present preliminary empirical findings **without providing more in-depth research**, such as theoretical analysis or effective methods for LLMs to address these shortcomings (e.g., how to overcome inconsistent performance and improve accuracy when relevant information is presented at a long distance).

3. Lack of discussion about preliminary research on LLMs' inconsistent performance related to graph representation [3].

4. The considered graph tasks are also limited. While the authors claim that Common Connection and Similarity are fundamental graph tasks, I struggle to identify direct application scenarios for these tasks. Additionally, existing research on LLMs solving basic graph reasoning tasks [1, 2] does not include these types, focusing instead on tasks like the shortest path. I believe that the shortest path is a type of graph task that requires specific pieces of information, i.e., the source and target nodes. Enriching this type of task could further highlight the inherent limitations of LLMs in solving graph problems.

---

[1] Chen, Nuo et al. "GraphWiz: An Instruction-Following Language Model for Graph Problems." In KDD, 2024.

[2] Luo, Zihan, et al. "GraphInstruct: Empowering Large Language Models with Graph Understanding and Reasoning Capability." In arXiv preprint, 2024.

[3] Ge, Yuyao, et al. "Can Graph Descriptive Order Affect Solving Graph Problems with LLMs?" In arXiv preprint, 2024.

**Questions:**

My primary concerns have been addressed in the **"Weaknesses"** section.  Minor questions:

* The most common representations of graphs are linked lists or adjacency dictionaries. I am confused about the "expert" encoding shown in Figure 1 and the partial explanation of this encoding in Line 131.

* What is the exact number of samples tested in Section 5.1? The authors only mention "hundreds of thousands of randomly generated graphs" in Line 311.

---

### Official Review · Reviewer_nfhA · 2024-11-03

**Soundness:** 1
**Presentation:** 1
**Contribution:** 2
**Rating:** 3
**Confidence:** 5

**Summary:**

The paper’s central claim is that LLMs’ performance on graph tasks that involves cross-referencing is influenced not only by absolute positioning of relevant information in the context (“lost-in-the-middle”), but by the relative distance between relevant information within the prompt. The authors term this phenomenon “lost-in-distance”. The authors first validate the lost-in-the-middle phenomenon in the graph setting using the Edge Existence task, and show that model performance decreases when relevant information is close to the middle of the prompt. The authors then address lost-in-distance using two more graph tasks: Common Connection and Similarity. The authors claim that results on the Common Connection task demonstrate models’ performance degradation as the distance between the relevant pieces of information in the two subgraphs increases. The authors also claim that a similar pattern of performance degradation can be seen in the Similarity task, where the model must compare connections across three nodes. Finally, the authors capture the effects of lost-in-distance by evaluating the goodness of fit of two proposed functions, comparing their ability to explain observed performance degradation.

**Strengths:**

- The concept of lost-in-distance is interesting, and is a significant-enough phenomenon worth exploring within the context of evaluating the reasoning abilities of LLMs.

**Weaknesses:**

=== Related Work and Conclusion ===
- The paper is incomplete due to the absence of related work and conclusion sections, which significantly limits the depth of analysis and hinders reader understanding. Without a related works section, readers are unable to properly contextualize the contributions of this paper relative to any prior work. The lack of a conclusion prevents readers from grasping the overall implications and the scope of potential future research directions.

=== Methodology & Analysis ===
- The crucial concept behind lost-in-distance is that model performance on cross-referencing tasks is impacted by the relative distance between relevant information in the prompt. However, this could apply to many other settings beyond graphs. If the authors show similar patterns in non-graph contexts, like in previously studied “needle-in-a-haystack” papers, it’ll strengthen their claims’ generalizability and impact.
- The authors mention that they “performed experiments on hundreds of thousands of randomly generated graphs to draw statistically significant conclusions regarding LLM behavior”. However, the exact number of graphs used in each experiment is unclear. The results in Figure 5 suggest that the number of graphs used in the Common Connection task is very small, raising concerns about the transparency and robustness of the results.
- For the Similarity task, the authors mention that their “findings indicate that when both distances are minimal, GPT4 and Llama-3-70B-Instruct exhibit the best performance.” However, Figure 12 contradicts this: in four out of six experiments, optimal performance was achieved at non-minimal distances (Small-Small). Therefore, this claim appears inconsistent with their results, and a direct comment/explanation on this from the authors would highly advisable.
- The authors demonstrate the high failure rate of Llama 3 8B, and therefore the analysis of these results, as well as the comparison of them to GPT-4 and Llama 3 70B, is highly limited. Including more SOTA LLMs that have lower failure rates, potentially including models in the Llama 3.1 family, would make the claims made by this paper, as well as inter-model comparison, stronger.
- The authors point out the limitations of models on these tasks, but do not attempt/suggest any potential improvements, such as in-context learning. On the subject of in-context learning, the authors mention that they “adopt Chain-of-Thought prompting (Wei et al., 2022) to guide the model in solving the task step by step”. According to Wei et al., 2022, CoT prompting involves placing in-context examples in the prompt with guided solutions. However, in Figure 11, the authors define CoT prompting as breaking the problem into subquestions. Why did the authors choose to define CoT prompting in this way? In addition, what was the baseline level of performance on the Similarity task without the use of subquestions?
- In Section A.2, the authors mention Repetition and Self-Contradiction as the two “most common patterns of degenerate answers”. It would be helpful to report how often both patterns occur, similarly to Table 4, as this would add clarity on the prevalence of these errors.

**Questions:**

- Does the calculation of the accuracies reported using Llama 3 8B take into account the high failure rate?
- Why did the authors choose to explore “lost-in-distance” only within the context of graphs?
- For each experiment, how many graphs did you evaluate on? How were the accuracies that the authors reported calculated?
- Why did the authors choose to define CoT prompting as decomposing the problem into subquestions instead of how CoT prompting is originally defined in Wei et al, 2022.?
- What was the baseline level of performance on the Similarity task without the use of subquestions?
- How often did both the Repetition and Self-Contradiction patterns occur?

---

### Official Review · Reviewer_1xBa · 2024-11-04

**Soundness:** 2
**Presentation:** 2
**Contribution:** 2
**Rating:** 3
**Confidence:** 5

**Summary:**

The paper investigates the limitations of large language models (LLMs) in processing graph-structured information when relevant context is distributed over larger distances within input sequences. The authors introduce the “lost-in-distance” phenomenon, demonstrating that LLM accuracy in graph tasks such as identifying common connections or assessing similarity between nodes declines significantly as the contextual distance between relevant nodes increases. Evaluations across LLMs reveal a universal decline in performance, independent of model size and graph encoding techniques, indicating inherent limitations in current LLM architectures for complex graph reasoning.

**Strengths:**

1.	The paper introduces the novel concept of "lost-in-distance," highlighting a previously unexamined limitation of LLMs in handling graph-structured information when relevant context is distributed over greater distances.
2.	By identifying the limitations of LLMs in processing complex, dispersed contextual information, the findings have broad implications for future LLM deployment in structured data scenarios.

**Weaknesses:**

1.	I believe the contributions of this work are limited.  The authors explore the impact of contextual positioning on LLMs within graph tasks.  However, numerous studies have already shown how context position and order affect LLMs' general generative capabilities in in-context learning (ICL).  The findings observed in this work appear consistent with these prior studies, simply applied to a specific task, and lack innovative setups or contributions.
2.	Furthermore, the authors focus on tasks like identifying common connections or assessing similarity between nodes, which limits the generality of the findings. A broader range of tasks should be considered. It’s worth noting that the effects of more general graph descriptive order on graph tasks have already been explored [1]. In contrast, this paper feels more like an experimental report on the impact of a specific descriptive order within a narrowly defined set of tasks in the graph domain.
[1] Ge Y et al. Can Graph Descriptive Order Affect Solving Graph Problems with LLMs?

**Questions:**

1. Given the identified “lost-in-distance” phenomenon, are there any potential strategies the authors would recommend for mitigating this effect?
2. The study uses multiple graph encoding techniques but does not deeply analyze the role of each encoding in mitigating or amplifying the “lost-in-distance” effect.  Could the authors elaborate on whether certain encoding styles are more effective in handling context separation?

---

### Meta-Review · Area_Chair_8GkW · 2024-12-16

**Metareview:**

Reviewers generally agree that the paper studies an important and novel problem, where the concept of 'lost-in-distance' provides useful insights in examining the limitations of LLMs.

However, the paper appears to be incomplete: lacking a thorough review of related work; no conclusion section; empirical results are limited to a narrowly defined set of tasks; the paper only presents the phenomenon, but lacks a discussion of potential measures or strategies to address it.

**Additional Comments On Reviewer Discussion:**

See summary of weaknesses above. The authors did not provide any reponse.

---

### Decision · Program_Chairs · 2025-01-22

Reject